# PATCH-LEVEL CONTRASTING WITHOUT PATCH CORRESPONDENCE FOR ACCURATE AND DENSE CONTRASTIVE REPRESENTATION LEARNING

**Shaofeng Zhang**[1,2], **Feng Zhu**[3], **Rui Zhao**[1,3], **Junchi Yan**[1,2*]
[1]MoE Key Lab of Artificial Intelligence, Shanghai Jiao Tong University
[2]Shanghai AI Laboratory    [3]SenseTime Group Limited
{sherrylone, yanjunchi}@sjtu.edu.cn
{zhufeng, zhaorui}@sensetime.com
Code:  https://github.com/Sherrylone/Query_Contrastive

## ABSTRACT

We propose ADCLR: Accurate and Dense Contrastive Representation Learning, a novel self-supervised learning framework for learning accurate and dense vision representation. To extract spatial-sensitive information, ADCLR introduces query patches for contrasting in addition with global contrasting. Compared with previous dense contrasting methods, ADCLR mainly enjoys three merits: i) achieving both global-discriminative and spatial-sensitive representation, ii) model-efficient (no extra parameters in addition to the global contrasting baseline), and iii) correspondence-free and thus simpler to implement. Our approach achieves new state-of-the-art performance for contrastive methods. On classification tasks, for ViT-S, ADCLR achieves 77.5% top-1 accuracy on ImageNet with linear probing, outperforming our baseline (DINO) without our devised techniques as plug-in, by 0.5%. For ViT-B, ADCLR achieves 79.8%, 84.0% accuracy on ImageNet by linear probing and finetune, outperforming iBOT by 0.3%, 0.2% accuracy. For dense tasks, on MS-COCO, ADCLR achieves significant improvements of 44.3% AP on object detection, 39.7% AP on instance segmentation, outperforming previous SOTA method SelfPatch by 2.2% and 1.2%, respectively. On ADE20K, ADCLR outperforms SelfPatch by 1.0% mIoU, 1.2% mAcc on the segmentation task.

## 1 INTRODUCTION

Self-supervised representation learning (SSL) has been attracting increasing attention for deep learning, whereby a prediction problem is often formulated by a pretext task for pre-training with unlabeled data. SSL methods can mainly be divided into three categories: **1) Generative** approaches (Goodfellow et al., 2014) learn to generate samples in the input space. However, generation can be computationally expensive and may not be necessary for representation learning. **2) Contextual** methods (Gidaris et al., 2018) design pretext tasks (denoising auto-encoders (Vincent et al., 2008), context auto encoders (Zhang et al., 2016), etc). **3) Contrastive** methods (Jin et al., 2022; Zhang et al., 2022; Chen et al., 2021; Caron et al., 2021) take augmented views of the same image as positive pairs and others as negative pairs. Contrastive-based methods have shown great promise e.g. in image classification/detection, video classification (Caron et al., 2021), and others (Chen et al., 2021).

It has been recently shown (Chen et al., 2020a; Wang et al., 2021) that existing contrastive learning in general aims to learn global-discriminative features, which may lack spatial sensitivity (Yi et al., 2022), and it limits their ability on downstream fine-tuning tasks, especially for dense vision tasks like detection and segmentation. Consequently, object-level (Wei et al., 2021; Hénaff et al., 2022) and pixel-level (Xie et al., 2021c; Wang et al., 2021) contrastive objectives and frameworks are proposed. Meanwhile, with the success of recent ViT-based visual backbones (Dosovitskiy et al., 2020; Liu et al., 2021), patch-level contrastive approaches (Yun et al., 2022) are devised, which

---

*Junchi Yan is the correspondence author. Rui Zhao is also with Qing Yuan Research Institute, Shanghai Jiao Tong University. This work was in part supported by NSFC (62222607), Shanghai Municipal Science and Technology Major Project (2021SHZDZX0102) and SenseTime Collaborative Research Grant.

achieve state-of-the-art performance on downstream dense tasks. However, there are mainly three disadvantages in these dense contrasting methods. **i)** It is hard to balance the global and patch-level losses in the dense contrasting methods (Xie et al., 2021c; Wang et al., 2021), causing their less competitive linear/fine-tune accuracy on the global classification task. **ii)** Establishing the correspondence among pixels/patches usually requires bilinear interpolation, which is complex and heavily sensitive to random crop augmentation (in an extreme case, if two views have no intersection parts, there's no correspondence relation). **iii)** each corresponding pixel/patch need be involved in the final contrastive loss, which is time-consuming. In this paper, we propose Accurate and Dense Contrastive Representation Learning (ADCLR), which is more global-discriminative, spatial-sensitive, correspondence-free and efficient. The main contributions of ADCLR are:

**1) Cross-view Query-based Patch-level Contrasting Paradigm:** For patch-level contrasting as recently used for dense tasks in negative-free Transformer-based SSL methods, we propose to augment two views, and perform contrasting crops from different views, for more effective learning with increased contrasting difficulty. The motivation for our cross-view design instead of the commonly-used single-view in existing patch-level contrasting is: it is non-triv and even impossible to establish patch correspondence within a single view (especially adding random resized crop augmentation). While introducing two views and replacing the correspondence establishing with more feasible query could meanwhile increase the patch appearance variance for more difficult contrasting. The above module can be introduced to existing global contrasting only SSL baselines. As shown in Fig. 1, the $[CLS]$ tokens are used to extract global information, and the designed query patches are used to extract local information, making ADCLR both global-discriminative and spatial-sensitive.

**2) Robust Unidirectional Cross-Attention Scheme under the above Paradigm:** The above patch-level contrasting paradigm can technically be prone to collapse to a trivial solution[1] (see more explanation in our theoretical analysis in Sec. 3.3) if we directly resort to the unidirectional self-attention scheme as used in the vanilla vision Transformers. Instead, we design unidirectional cross-attention, which takes both query patches and raw patches from raw images as input. For attention block, the data flow are: $RP \to RP, \{RP, QP_i\} \to QP_i, QP_i \nrightarrow QP_j \ (i \neq j)$ where $RP$ and $QP_i$ means raw patches (including $[CLS]$ token) and the $i$-th query patch, respectively.

**3) Boosting baselines to new SOTA accuracy on classification and dense tasks:** The proposed ADCLR can serve as a plugin based on existing Transformer-based and negative-free SSL baselines e.g. DINO (Caron et al., 2021) and iBOT (Zhou et al., 2022). Our experimental results on both linear probing, finetune classification as well as other downstream tasks show the effectiveness of ADCLR.

## 2 RELATED WORKS

In this section, we review previous literature on general contrastive learning (for global tasks such as image classification) and dense ones (for downstream tasks like detection and segmentation).

**General contrastive learning** aims at learning the global information of images through enforcing contrastive objectives (e.g., InfoNCE loss, Hjelm et al. (2018)) on the final global image representations ($[CLS]$ token or from average pooling). The pivotal idea is to align the embeddings of two augmented views from the same image while preventing trivial solutions (a.k.a. degenerated representations). To reach this target, MoCo (He et al., 2020; Chen et al., 2020b) employs a memory bank to store and update negative examples, from which the negative examples could be randomly sampled, whereas SimCLR (Chen et al., 2020a) treats all other data samples within the same training batch as negative examples. Due to the inconvenience and huge cost of using negative examples, later research turned to explore negative-example-free methods. BYOL (Grill et al., 2020) and SimSiam (Chen & He, 2021) design asymmetric architectures with additional predictors and stop-gradient to prevent using negative examples. Barlow Twins (Zbontar et al., 2021), ZeroCL (Zhang et al., 2021) and VICReg (Bardes et al., 2022) resort to feature-level decorrelation to mitigate trivial solutions. Inspired by the success of Vision Transformers (ViT), CNN backbones are gradually replaced with ViTs Chen et al. (2021); Caron et al. (2021). iBOT (Zhou et al., 2022) further incorporates contrastive learning with masked image modeling and has achieved State-of-the-Art performance on various tasks.

**Dense contrastive learning**, on the contrary, targets learning local information through regularizing different local regions of images. One common dense contrastive learning way is to mine the correspondence of each pixel (CNNs) or patch (ViTs) in a feature map (Wang et al., 2021; Yun et al.,

---

[1]The embedding vectors end up spanning a lower-dimensional subspace instead of the entire space.

2022). DenseCL (Wang et al., 2021) exploits the correspondence through sorting the similarities of pixels, while PixPro (Xie et al., 2021c) utilizes the augmentation wrapper to get the spatial correspondence of the pixel intersection between two views in the feature map. Furthermore, Detco (Xie et al., 2021a) tries to improve the performance of general contrastive learning approaches by augmenting multiple global and local views simultaneously. Inspired by PixPro, Resim (Xiao et al., 2021) uses Precise RoI Pooling (Jiang et al., 2018) to extract a feature vector from the associated feature map region for both views. On the basis of DenseCL, SetSim (Wang et al., 2022) employs a threshold selection to filter out noisy backgrounds. With the development of ViT in SSL (Chen et al., 2021; Caron et al., 2021), SelfPatch (Yun et al., 2022) treats the spatial neighbors of the patch as positive examples for learning more semantically meaningful relations among patches.

## 3 METHODOLOGY

We first review recent SSL approaches with ViTs as backbones (Caron et al., 2021; Yun et al., 2022) in Sec. 3.1. Then we present ADCLR detailedly in Sec. 3.2 (we provide a sketch in Fig. 1).

### 3.1 PRELIMINARIES

**Vision Transformers.** Denote an image by $\mathbf{x} \in \mathbb{R}^{C \times H \times W}$, where $H \times W$ is the resolution of the image and $C$ is the number of channels. Plain ViT (Dosovitskiy et al., 2020) treats the image $\mathbf{x}$ as a sequence composed of non-overlapping patches $\{\mathbf{x}^{(i)} \in \mathbb{R}^{CP^2}\}_{i=1}^{N}$, where each patch has a fixed $P \times P$ resolution. Then, the patches are linearly transformed to $D$-dimensional patch embeddings $\mathbf{z}^{(i)} = \mathbf{E}\mathbf{x}^{(i)} + \mathbf{W}_{pos}^{i} \in \mathbb{R}^{D}$, where $\mathbf{E} \in \mathbb{R}^{D \times CP^2}$ is the linear projection and $\mathbf{W}_{pos} \in \mathbb{R}^{D}$ is the positional embedding for the $i$-th patch. A $[CLS]$ token $\mathbf{z}^{[CLS]} \in \mathbb{R}^{D}$ is subsequently prepended to the patch sequence to extract global information, so the resulting input sequence is represented as $\mathbf{z} = [\mathbf{z}^{[CLS]}, \mathbf{z}^{(1)}, \mathbf{z}^{(2)}, \cdots, \mathbf{z}^{(N)}]$. Then, ViT uses a Transformer encoder (Vaswani et al., 2017) to generate both image-level ($[CLS]$ token) and patch-level (other tokens) . In line with SelfPatch (Yun et al., 2022), we use $f_\theta$ to denote the whole process of a ViT parameterized by $\theta$:

$$f_\theta(\mathbf{x}) = f_\theta\left(\left[\mathbf{z}^{[CLS]}, \mathbf{z}^{(1)}, \mathbf{z}^{(2)}, \cdots, \mathbf{z}^{(N)}\right]\right) = \left[f_\theta^{[CLS]}(\mathbf{x}), f_\theta^{(1)}(\mathbf{x}), f_\theta^{(2)}(\mathbf{x}), \cdots, f_\theta^{(N)}(\mathbf{x})\right],$$
(1)

where $f_\theta^{[CLS]}(x)$ and $f_\theta^{(i)}(\mathbf{x})$ are the representations of the whole image and $i$-th patch, respectively.

**Self-supervised learning with ViTs.** Since our ADCLR is built on top of DINO and iBOT, we shortly review DINO's framework and objective function. Given the image $\mathbf{x}$, DINO constructs a positive pair $(\mathbf{x}_A, \mathbf{x}_B)$ through random augmentation. Then, DINO learns by maximizing the similarity between their representations. The generic form of image-level self-supervised loss is:

$$\mathcal{L}_{DINO} = H\left(g_\gamma\left(f_\theta^{[CLS]}(\mathbf{x}_A)\right), sg\left(g_{\gamma'}(f_{\theta'}^{[CLS]}(\mathbf{x}_B))\right)\right),$$
(2)

where $H(a, b) = -a \log b$ is the cross entropy loss. $sg(\cdot)$ means stop-gradient operation. $g_\gamma$ is the MLP projector, which is commonly used in previous SSL methods (Chen et al., 2020a; Grill et al., 2020). $\gamma'$ and $\theta'$ denote the exponential moving averages updated parameters in the teacher branch.

### 3.2 ADCLR: ACCURATE AND DENSE CONTRASTIVE REPRESENTATION LEARNING

**Query crops.** Although DINO is able to handle most cases (e.g., two augmented views contain the same objects), we find that the $[CLS]$ token may extract mismatched information (see the attention map of $[CLS]$ in Fig. 1). To this end, we randomly crop each image to generate $Q$ query crops, where $Q$ is a pre-defined hyper-parameter. Then, we resize each query crop to the same resolution as raw patches (e.g., $16 \times 16$ or $8 \times 8$): $\mathbf{X}^q \in \mathbb{R}^{Q \times CP^2}$. Denote the $i$-th query crop as $\mathbf{x}^{q_i}$, we feed each query patch into a linear projector to get its embedding $\mathbf{z}^{(q_i)}$. Different from raw patches, it's **unnecessary** to add positional embeddings to query patches as the region of the query crop may not in the augmented views. Then, the input embeddings can be formulated as:

$$\mathbf{z} = \left[\mathbf{z}^{[CLS]}, \underbrace{\mathbf{z}^{(1)}, \mathbf{z}^{(2)}, \cdots, \mathbf{z}^{(N)}}_{\text{raw patches}}, \underbrace{\mathbf{z}^{(q_1)}, \mathbf{z}^{(q_2)}, \cdots, \mathbf{z}^{(q_Q)}}_{\text{query patches}}\right].$$
(3)

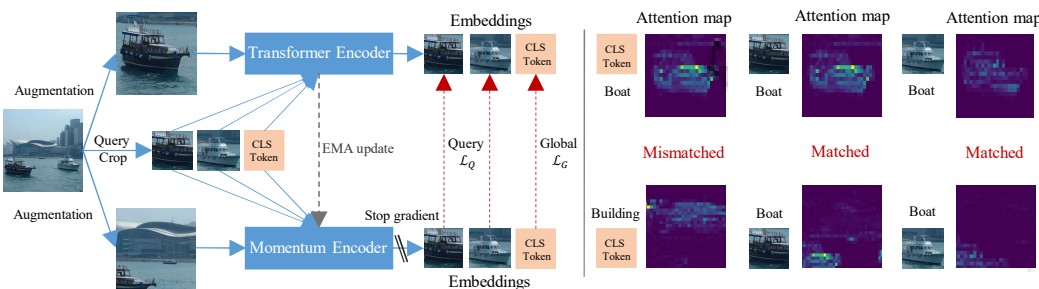

Figure 1: Framework of the proposed ADCLR (left figure). Here we set the number of patches set as 2 for better illustration. For each query patch, we add the contrastive loss $\mathcal{L}_Q$. The stop-gradient and momentum update operations are added to prevent collapse. The right figure illustrates the average attention map of all the heads in the last block. "Matched" means objects are from the same classes.

Note that $\mathbf{z}^{[CLS]}$ and raw patches $[\mathbf{z}^{(1)}, \mathbf{z}^{(2)}, \cdots, \mathbf{z}^{(N)}]$ will be equipped with positional embeddings.

**Unidirectional cross attention mechanism.** Instead of directly feeding the whole sequence (namely $[CLS]$ token, raw patches, query patches) to the Transformer encoder, which may hurt the performance of the downstream task (as only $[CLS]$ token and raw patches are fed to the Transformer for downstream tasks), we slightly modify the attention mechanism. Given one $\mathbf{z}^{[CLS]}$ token, raw patches sequence $[\mathbf{z}^{(1)}, \mathbf{z}^{(2)}, \cdots, \mathbf{z}^{(N)}]$ and query patches sequence $[\mathbf{z}^{(q_1)}, \mathbf{z}^{(q_2)}, \cdots, \mathbf{z}^{(q_Q)}]$. Let $\mathbf{z}' = [\mathbf{z}^{[CLS]}, \mathbf{z}^{(1)}, \mathbf{z}^{(2)}, \cdots, \mathbf{z}^{(N)}]$ and $\mathbf{z}^q = [\mathbf{z}^{(q_1)}, \mathbf{z}^{(q_2)}, \cdots, \mathbf{z}^{(q_Q)}]$. For $\mathbf{z}^{[CLS]}$ and raw patches, the propagation rule in the attention block is in line with plain Transformer, i.e.,

$$\text{Attn}(\mathbf{Q}_{\mathbf{z}'}, \mathbf{K}_{\mathbf{z}'}, \mathbf{V}_{\mathbf{z}'}) = \text{Softmax}\left(\frac{\mathbf{Q}_{\mathbf{z}'}\mathbf{K}_{\mathbf{z}'}^\top}{\sqrt{d_k}}\right)\mathbf{V}_{\mathbf{z}'}, \qquad (4)$$

where $\mathbf{Q}_{\mathbf{z}'} = \mathbf{z}'\mathbf{W}_{\mathbf{Q}}, \mathbf{K} = \mathbf{z}'\mathbf{W}_{\mathbf{K}}, \mathbf{V} = \mathbf{z}'\mathbf{W}_{\mathbf{V}}$, and $\mathbf{W}_{\mathbf{Q}}, \mathbf{W}_{\mathbf{K}}, \mathbf{W}_{\mathbf{V}}$ are learnable weights. For query patches $\mathbf{z}^{(q_i)}$, the attention propagation rule is similar to $[CLS]$ token, i.e.,

$$\text{Attn}(\mathbf{Q}_{\mathbf{z}^{(q_i)}}, \mathbf{K}_{\mathbf{z}'}, \mathbf{V}_{\mathbf{z}'}) = \text{Softmax}\left(\frac{\mathbf{Q}_{\mathbf{z}^{(q_i)}}\mathbf{K}_{\mathbf{z}'}^\top}{\sqrt{d_k}}\right)\mathbf{V}_{\mathbf{z}'}, \quad 1 \le i \le Q, \qquad (5)$$

where $\mathbf{Q}_{\mathbf{z}^{(q_i)}} = \mathbf{z}^{(q_i)}\mathbf{W}_{\mathbf{Q}}$. By Eq. 4 and Eq. 5, we can complete the attention block in modified Transformer. In a nutshell, we can summarize the propagation rule as $RP \to RP, \{RP, QP_i\} \to QP_i, QP_i \nrightarrow QP_j \ (i \ne j)$, where $RP$ and $QP_i$ means raw patches (including $[CLS]$ token) and $i$-th query patch, respectively. We can find only raw patches and $[CLS]$ token linearly express the raw patches and $[CLS]$ in the attention mechanism. This design can make the pretrained Transformer more suitable for downstream tasks without introducing extra parameters.

**Objective function of ADCLR.** By feeding the whole sequence of two views $\mathbf{x}_A$ and $\mathbf{x}_B$ to the modified Transformer $f_\theta$ and MLP projection head $g_\gamma$, we can obtain the final representation of the image $\mathbf{h}_{A;B}^{[CLS]}$, raw patches $\mathbf{h}_{A;B}$ and query patches $\mathbf{h}_{A;B}^q$. Then, the overall objectives are:

$$\mathcal{L}_{ADCLR} = \underbrace{H(\mathbf{h}_A^{[CLS]}, \mathbf{h}_B^{[CLS]})}_{\text{Global}} + \underbrace{\frac{\lambda}{Q}\sum_{i=1}^{Q} H(\mathbf{h}_A^{(q_i)}, \mathbf{h}_B^{(q_i)})}_{\text{Local}}, \qquad (6)$$

where $\lambda$ is used to balance global and local loss. The impact of $\lambda$ is studied in Sec. 4.3. ADCLR can also selectively add the MIM objective in iBOT Zhou et al. (2022) to further boost the accuracy.

### 3.3 NECESSITY OF OUR UNIDIRECTIONAL CROSS-ATTENTION MECHANISM

**Theorem 1** *Bidirectional self-attention in standard vision Transformer leads to collapse. Given a set of query patches $\{\mathbf{x}\}_{i=1}^Q$ and a set of raw patches $\{\mathbf{x}\}_{i=1}^N$, contrasting with the standard vision Transformer-like backbones that use **bidirectional self-attention mechanisms** can cause ADCLR easily to collapse if $Q$ is large enough ($Q >>$ the total number of raw patches of the image).*

Table 1: Linear and finetune Top-1 accuracy on ImageNet-1K. † means using DALL·E (Reddy et al., 2021) tokenizer in the pre-training stage. ‡ means these works are so far arXiv preprints to date. * means adding the mask image modeling loss proposed in (Zhou et al., 2022)

| Framework | Method | Model | #Params | PT Eps. | Linear | FT Eps. | FT Acc. (%) |
|---|---|---|---|---|---|---|---|
| Training from scrach | Scratch, DeiT | ViT-B | 86M | 0 | - | 300 | 81.8 |
| | Scratch, MAE | ViT-B | 86M | 0 | - | 300 | 82.3 |
| | Scratch, Swin | Swin-B | 88M | 0 | - | 300 | 83.5 |
| Supervised Pre-training | Supervised, SimMIM | Swin-B | 88M | 300 | - | 100 | 83.3 |
| | Supervised, SimMIM | Swin-L | 197M | 300 | - | 100 | 83.5 |
| Masked Image Modeling | MAE | ViT-B | 86M | 1600 | 67.8 | 100 | 83.6 |
| | BEiT† | ViT-B | 86M | 300 | 37.6 | 100 | 83.0 |
| | CAE†,‡ | ViT-B | 86M | 300 | 64.2 | 100 | 83.3 |
| | CAE†,‡ | ViT-B | 86M | 800 | 68.3 | 100 | 83.6 |
| | CIM‡ | ViT-B | 86M | 300 | - | 100 | 83.3 |
| Contrastive Leanring | Moco V3 | ViT-B | 86M | 800 | 76.5 | 100 | 83.2 |
| | DINO | ViT-B | 86M | 400 | 78.2 | 100 | 83.6 |
| | iBOT | ViT-B | 86M | 400 | 79.5 | 100 | 83.8 |
| | ADCLR | ViT-B | 86M | 400 | **78.6** | 100 | **83.9** |
| | ADCLR* | ViT-B | 86M | 400 | **79.8** | 100 | **84.0** |

**Remarks.** In fact, applying the standard bidirectional self-attention as used in vision Transformers not only incurs the solution collapse issue as shown in Theorem 1, but also may hurt the performance, as will be shown in the ablation studies in Sec. 4.3. We guess this is because ADCLR introduces additional query patches during the pretraining stage, while the downstream task finetuning only involves raw patches and the bidiretional scheme would cause the mutual influence between query and raw patches, leading to the input inconsistency between pretraining and downstream tasks. Therefore, we elaborately replace the bidirectional self-attention scheme by our devised unidirectional cross-attention one as such the above two issues are addressed directly.

## 4 EXPERIMENTS

### 4.1 EXPERIMENT SETUP

**Dataset.** We conduct self-supervised pre-training on the ImageNet-1K (Deng et al., 2009) training set, as commonly used in SSL methods for both MIM (He et al., 2021) and contrastive learning (Chen et al., 2020a). ImageNet-1k includes 1,000 classes which are even in distribution. We also transfer the encoder pretrained by ADCLR on MS-COCO (Lin et al., 2014), ADE20K (Zhou et al., 2017) and other fine-grained classification datasets (Van Horn et al., 2018) for downstream evaluation.

**Baselines.** We consider a variety of existing SSL methods based on the ResNet (He et al., 2016) and ViT (Dosovitskiy et al., 2020) architectures: (a) self-supervised ResNets: MoCo-v2 (Chen et al., 2020b), SwAV (Caron et al., 2020), Barlow Twins (Zbontar et al., 2021), ZeroCL (Zhang et al., 2021), ARB (Zhang et al., 2022), DenseCL (Wang et al., 2021), ReSim (Xiao et al., 2021), and DetCo (Xie et al., 2021a); and (b) self-supervised ViTs: DINO (Caron et al., 2021), MoCo-v3 (Chen et al., 2021), MoBY (Xie et al., 2021b), iBOT (Zhou et al., 2022) and SelfPatch (Yun et al., 2022).

**Evaluation protocols.** Standard SSL protocols is to either learn a linear classifier on frozen features (Chen et al., 2020a; He et al., 2020) or to finetune on downstream tasks (He et al., 2021; Chen et al., 2022). For linear evaluations, we apply random resize crops and horizontal flips augmentation for training, and report accuracy on a central crop. For finetuning evaluations (detection and segmentation on MS-COCO (Lin et al., 2014), segmentation on ADE20K (Zhou et al., 2017)), we initialize networks with the pretrained weights to adapt with further training. In line with (Zbontar et al., 2021), we also evaluate our method's transfer ability on small-scale and fine-grained classification dataset.

### 4.2 MAIN RESULTS

We first validate the ADCLR framework used in this study with the standard self-supervised benchmark on ImageNet. Then, we study the performance of detection, segmentation, and transfer learning.

**Finetune on ImageNet-1k.** Table 1 shows the results with four different learning schemes, i.e., finetune from scratch, supervised pre-training, MIM-based approaches, and contrastive learning. We report linear probing and finetune accuracy, respectively. Our finetune protocol is in line with iBOT (Zhou et al., 2022), using strong regularization for training 100 epochs. The compared baselines

Table 2: $k$-NN and linear top-1 accuracy on ImageNet-1k. Epoch$^\dagger$ means effective pre-training epochs accounting for actual trained images/views, which is used in iBOT (Zhou et al., 2022). * means using MIM loss and #Views means the number of views used in the pretraining stage.

| | Method | Model | #Params. | images/s | #Views | Epoch$^\dagger$ | $k$-NN | Linear |
|---|---|---|---|---|---|---|---|---|
| | Supervised | RN50 | 23M | 1237 | 1 | 300 | 79.3 | 79.3 |
| | SimCLR (Chen et al., 2020a) | RN50 | 23M | 1237 | 2 | 1000 | 60.1 | 69.1 |
| | MocoV2 (Chen et al., 2020b) | RN50 | 23M | 1237 | 2 | 800 | 61.9 | 71.1 |
| | ARB (Zhang et al., 2022) | RN50 | 23M | 1237 | 2 | 1000 | 66.4 | 73.1 |
| CNNs | Barlow Twins (Zbontar et al., 2021) | RN50 | 23M | 1237 | 2 | 1000 | 66.0 | 73.2 |
| | BYOL (Grill et al., 2020) | RN50 | 23M | 1237 | 2 | 1000 | 64.8 | 74.4 |
| | SwAV (Caron et al., 2020) | RN50 | 23M | 1237 | 8 | 800 | 65.7 | 75.3 |
| | DINO (Caron et al., 2021) | RN50 | 23M | 1237 | 12 | 3200 | 67.5 | 75.3 |
| | Moco V3 (Chen et al., 2021) | ViT-S/16 | 22M | 1007 | 2 | 1200 | - | 73.4 |
| | Moco V3 (Chen et al., 2021) | ViT-B/16 | 86M | 312 | 2 | 1200 | - | 76.7 |
| | SwAV (Caron et al., 2020) | ViT-S/16 | 22M | 1007 | 8 | 2400 | 66.3 | 73.5 |
| | DINO (Caron et al., 2021) | ViT-S/16 | 22M | 1007 | 12 | 3200 | 74.5 | 77.0 |
| | DINO (Caron et al., 2021) | ViT-B/16 | 86M | 312 | 12 | 1600 | 76.1 | 78.2 |
| ViTs | iBOT (Zhou et al., 2022) | ViT-S/16 | 22M | 1007 | 12 | 3200 | 75.2 | 77.9 |
| | iBOT (Zhou et al., 2022) | ViT-B/16 | 86M | 312 | 12 | 1600 | 77.1 | 79.5 |
| | ADCLR | ViT-S/16 | 22M | 1007 | 12 | 3200 | 74.6 | **77.5** |
| | ADCLR | ViT-B/16 | 86M | 312 | 12 | 1600 | 76.6 | **78.6** |
| | ADCLR* | ViT-S/16 | 22M | 1007 | 12 | 3200 | 74.9 | **78.1** |
| | ADCLR* | ViT-B/16 | 86M | 312 | 12 | 1600 | 77.4 | **79.8** |

Table 3: Object detection and instance segmentation on MS-COCO. Mask R-CNN is adopted and trained with the 1x schedule. All the results are based on the same implementation for object detection and instance segmentation. Epoch refers to the number of pretraining epochs on ImageNet-1K.

| Method | Backbone | Epoch | #Param. | Detection | | | Segmentation | | |
|---|---|---|---|---|---|---|---|---|---|
| | | | | $AP^{bb}$ | $AP^{bb}_{50}$ | $AP^{bb}_{75}$ | $AP^{mk}$ | $AP^{mk}_{50}$ | $AP^{mk}_{75}$ |
| Moco-V2 (Chen et al., 2020b) | RN50 | 200 | 23M | 38.9 | 59.2 | 42.4 | 35.5 | 56.2 | 37.8 |
| SwAV (Caron et al., 2020) | RN50 | 200 | 23M | 38.5 | 60.4 | 41.4 | 35.4 | 57.0 | 37.7 |
| DenseCL (Wang et al., 2021) | RN50 | 200 | 23M | 40.3 | 59.9 | 44.3 | 36.4 | 57.0 | 39.2 |
| ReSim (Xiao et al., 2021) | RN50 | 200 | 23M | 40.3 | 60.6 | 44.2 | 36.4 | 57.5 | 38.9 |
| DetCo (Xie et al., 2021a) | RN50 | 200 | 23M | 40.1 | 61.0 | 43.9 | 36.4 | 58.0 | 38.9 |
| Moco V3 (Chen et al., 2021) | ViT-S/16 | 300 | 23M | 39.8 | 62.6 | 43.1 | 37.1 | 59.6 | 39.2 |
| MoBY (Xie et al., 2021b) | ViT-S/16 | 300 | 22M | 41.1 | 63.7 | 44.8 | 37.3 | 60.3 | 39.8 |
| DINO (Caron et al., 2021) | ViT-S/16 | 300 | 22M | 40.8 | 63.4 | 44.2 | 37.3 | 59.9 | 39.5 |
| SelfPatch (Yun et al., 2022) | ViT-S/16 | 200 | 22M | 42.1 | 64.9 | 46.1 | 38.5 | 61.3 | 40.8 |
| ADCLR | ViT-S/16 | 200 | 22M | 43.8 | 65.2 | 47.4 | 39.2 | 61.9 | 41.3 |
| ADCLR | ViT-S/16 | 300 | 22M | **44.3** | **65.4** | **47.6** | **39.7** | **62.1** | **41.5** |

are MIM-based methods MAE (He et al., 2021), BEiT (Bao et al., 2021), CIM (Fang et al., 2022) as well as contrastive methods (Caron et al., 2021; Zhou et al., 2022; Chen et al., 2021) and the combination of the two techniques: CAE (Chen et al., 2022) which is emerging.

**Linear probing on ImageNet-1k.** For linear probing evaluation, we compare ADCLR with both CNNs-based and ViTs-based contrastive learning methods. Following iBOT (Zhou et al., 2022), we use effective training epochs to measure the images actually seen by ADCLR. Specifically, DINO and iBOT are by default trained with 2 global crops with the crop size 224*224 and 10 local crops with the size 96*96. The calculation formula is $r = 2 + (\frac{96}{224})^2 \times 10 = 3.84 \approx 4$. Moreover, ADCLR introduces query crops, and the total seen image is $r = 2 + (\frac{96}{224})^2 \times 10 + (\frac{16}{224})^2 \times 10 = 3.89 \approx 4$. The overall seen image brought by query crops is only 0.05, which can be negligible. We report $k$-NN and linear probing accuracy in Table 2. ADCLR achieves **77.5%** top-1 accuracy on ImageNet-1k with ViT-S, outperforming baseline DINO **+0.5%** in 3200 effective epochs. For ViT-B, ADCLR outperforms DINO and iBOT **+0.4%** and **+0.3%** accuracy in 800 effective epochs, respectively.

#### 4.2.1 TRANSFER LEARNING ACROSS DIFFERENT TASKS

**MS-COCO Setup.** We evaluate pre-trained models on the MS-COCO object detection and instance segmentation tasks. Here, all models are fine-tuned with Mask R-CNN (He et al., 2017) and FPN (Lin et al., 2017) under the standard 1x schedule. Following (Chen et al., 2022), we utilize multi-scale training and resize the image with the size of the short side between 480 and 800 and the long side no larger than 1333. In pretraining stage, we set $\lambda = 0.5$ and $Q = 10$ for extract better dense information. The pretrining is in line with SelfPatch (Yun et al., 2022). In finetune stage, we distributed training

on 8 Tesla V100 32G GPUs set batch size 32 and set the learning rate as 3e-4 with the layer decay rate 0.75, which also follows SelfPatch.

**Results.** Table 3 shows the detection and segmentation performance on MS-COCO. We give the results of both 200 and 300 epochs' pretraining to better demonstrate the advance of ADCLR. We find that Although both SelfPatch and ADCLR perform patch-level contrastive loss, ADCLR breaks the limitation of contrasting between the same view. The learning paradigm of query tokens seems to bring more variance than single view contrasting, increasing the difficulty of patch-level pretext tasks, and resulting in higher performance on detection and segmentation tasks. Specifically, under 200 epochs pretraining, ADCLR outperforms SelfPatch with **+1.7%** and **+0.7%** points on detection ($AP^{bb}$) and segmentation ($AP^{mk}$) tasks, respectively. Compared with baseline DINO (Caron et al., 2021), ADCLR outperforms DINO with a large range, e.g., **+2.2%** bounding box average precision, **+1.2%** mask average precision higher than DINO, which benefits from contrasting on query patches.

Table 4: **ADE20K semantic segmentation** performances of the recent self-supervised approaches pre-trained on ImageNet. The metrics mIoU, aAcc, and mAcc denote the mean intersection of union, all pixel accuracy, and mean class accuracy, respectively.

| Method | Arch | Backbone | #Iter. | mIoU | aAcc | mAcc |
|---|---|---|---|---|---|---|
| MoCo-v2 (Chen et al., 2020b) | FPN | ResNet50 | 40k | 35.8 | 77.6 | 45.1 |
| SwAV (Caron et al., 2020) | FPN | ResNet50 | 40k | 35.4 | 77.5 | 44.9 |
| DenseCL (Wang et al., 2021) | FPN | ResNet50 | 40k | 37.2 | 78.5 | 47.1 |
| MocoV3 (Chen et al., 2021) | FPN | ViT-S/16 | 40k | 35.3 | 78.9 | 45.9 |
| MoBY (Xie et al., 2021b) | FPN | ViT-S/16 | 40k | 39.5 | 79.9 | 50.5 |
| DINO (Caron et al., 2021) | FPN | ViT-S/16 | 40k | 38.3 | 79.0 | 49.4 |
| DINO (Caron et al., 2021) | UperNet | ViT-S/16 | 160k | 42.3 | 80.4 | 52.7 |
| SelfPatch (Yun et al., 2022) | FPN | ViT-S/16 | 40k | 41.2 | 80.7 | 52.1 |
| SelfPatch (Yun et al., 2022) | UperNet | ViT-S/16 | 160k | 43.2 | 81.5 | 53.9 |
| ADCLR | FPN | ViT-S/16 | 40k | 42.4 | 81.1 | 54.2 |
| ADCLR | UperNet | ViT-S/16 | 160k | **44.2** | **81.8** | **55.1** |

**Semantic segmentation on ADE20K. Setups.** We also evaluate semantic segmentation performances of pre-trained models on ADE20K (Zhou et al., 2017), which includes 150 fine-grained semantic categories and 25k training data. In line with SelfPatch (Yun et al., 2022), we report three metrics: the mean intersection of union (mIoU) averaged over all semantic categories, all pixel accuracy (aAcc), and mean class accuracy (mAcc). We pretrain ADCLR in 200 epochs and finetune the pretrained backbone in different architectures (FPN (Lin et al., 2017) and UperNet (Xiao et al., 2018)) for fair comparisons. **Results.** The results are given in Table 4. When finetuning the FPN with 40k iterations, ADCLR outperforms SelfPatch and DINO **+1.2%** and **+3.9%** mIoU points, respectively. In UperNet with 160k iterations, ADCLR outperforms SelfPatch and DINO **+1.0%** and **+3.0%** mIoU points. We guess the gain mainly comes from two aspects, 1) patch-level contrasting enhances the ability to capture local information (see the gap between global method DINO and local method SelfPatch and ADCLR); 2) The learning paradigm of query tokens brings more accurate information (the gap between SelfPatch and ADCLR).

**Transfer learning on the small-scale dataset.** We study transfer learning where we pre-train on ImageNet-1K and fine-tune on several smaller datasets. We follow the training recipe and protocol used in DINO and iBOT. For pretraining, ViT-S and ViT-B are pretrained on 16 GPUs with 1024 batch size and 10 local views in 800 and 400 epochs, respectively. The results are reported in Table 5. While the results on several datasets (e.g., CIFAR10, CIFAR100, Flowers, and Cars) have almost plateaued, ADCLR consistently performs favorably against other SSL frameworks, achieving state-of-the-art transfer results. Specifically, on small-scale datasets, ADCLR outperforms baseline DINO (Caron et al., 2021) over 2.2% and 1.1% accuracy on iNaturalist18 and iNaturalist19 datasets. We observe ADCLR achieve greater performance gain over DINO in fine-grained classification datasets like iNaturalist18, iNaturalist19, and Flowers than CIFAR10 and CIFAR100, indicating ADCLR (query token) is more adept at capturing local information. Compared with the state-of-the-art method iBOT (Zhou et al., 2022), although ADCLR achieves similar results, iBOT spends more memory due to the MIM branch (under the same setting, 19.5G for iBOT and 16.7G for ADCLR).

## 4.3 ABLATION STUDY

To verify the robustness of ADCLR, we comprehensively study the sensitiveness of balance ratio $\lambda$, number of query tokens $Q$, the effect of cross attention mechanism, and crop ratio.

Table 5: Top-1 linear accuracy on transfer learning, all models are pretrained on ImageNet-1K.

| Arch. | Method | CIFAR10 | CIFAR100 | iNaturalist18 | iNaturalist19 | Flowers | Cars |
|-------|--------|---------|----------|---------------|---------------|---------|------|
| | Rand. | 99.0 | 89.5 | 70.7 | 76.6 | 98.2 | 92.1 |
| | BEiT (Bao et al., 2021) | 98.6 | 87.4 | 68.5 | 76.5 | 96.4 | 92.1 |
| ViT-S | iBOT (Zhou et al., 2022) | **99.1** | 90.7 | 73.7 | **78.5** | **98.6** | 94.0 |
| | DINO (Caron et al., 2021) | 99.0 | 90.5 | 72.0 | 78.2 | 98.5 | 93.0 |
| | ADCLR | **99.1** | **90.8** | **74.0** | **78.5** | **98.6** | **94.2** |
| | Rand. | 99.0 | 90.8 | 70.7 | 76.6 | 98.2 | 92.1 |
| | BEiT (Bao et al., 2021) | 99.0 | 90.1 | 72.3 | 79.2 | 98.0 | 94.2 |
| ViT-B | iBOT (Zhou et al., 2022) | **99.2** | **92.2** | 74.6 | 79.6 | **98.9** | 94.3 |
| | DINO (Caron et al., 2021) | 99.1 | 91.7 | 72.6 | 78.6 | 98.8 | 93.0 |
| | ADCLR | **99.2** | 92.0 | **74.8** | **79.7** | **98.9** | **94.6** |

(a) Effect of $\lambda$     (b) Effect of query crops $Q$     (c) Uni-/Bi- directional attention

Figure 2: Top-1 linear classification accuracy on ImageNet-1K with different hyper-parameters.

**Effect of $\lambda$.** We conduct ablation experiments on different balance ratios $\lambda$ to explore the effect of local objective $\mathcal{L}_{Local}$. Specifically, we set batch size as 1024, and distribute training on 32 GPUs with 30 epochs as warm up out of the total 100 epochs. For efficiency, we only use two global views (224 * 224), and 10 query views (16*16) without local view (96*96). We report the results in Fig. 2(a). Compared with baseline DINO, ADCLR outperforms **+1.3%** top-1 accuracy when $\lambda = 0.5$. With too large $\lambda$, the accuracy will drop a little, which we guess ADCLR would overemphasize the local information, but ignore the global information and leading to the decreasing accuracy of global task (classification). With smaller $\lambda$ e.g. between 0.1 and 0.2, ADCLR is mainly determined by $[CLS]$ token, which may bring unwanted contrasting loss (see Fig. 1) to hurt the final performance.

**Effect of the number of query crops $Q$.** We set batch size as 1024 and pretrain ADCLR with 100 epochs. In line with ablation on $\lambda$, we only use two global views without local views. Besides, we choose the best value of $\lambda = 0.5$ for this experiment. We illustrate the results in Fig. 2(b). The performance increases with the increase of $Q$, which is similar to the ensemble mechanism (with more base models, the ensemble model are more accurate). We find ADCLR saturates when $Q = 9$ and $Q = 10$, which may be enough to capture local information. We also try large $Q$, but when we set $Q = 15$, we find there's only little improvement, and thus we set $Q = 10$ by default.

Table 6: Ablation study query crop ratio.

| ViT-S, 100 epochs | 0.1 | 0.15 | 0.2 |
|-------------------|-----|------|-----|
| ADCLR (Linear) | 68.5 | **69.1** | 68.8 |

**Effect of unidirectional cross attention.** As discussed in Sec. 3.2, we find unidirectional cross attention makes ADCLR more effective to downstream tasks (with only the raw patches and $[CLS]$ tokens as input). We conduct experiments by using unidirectional and bidirectional cross attention mechanisms and the results are given in Fig. 2(c). When $Q = 1$, ADCLR (bidir) drops about 3%~4% accuracy. When increasing the number of query view $Q$, the accuracy of the downstream classification task drops notably. When we set $Q = 196$ (same with query token), we find ADCLR (bidir) only gets about 50% accuracy, while ADCLR (unidir) gets 68.8% accuracy.

**Effect of query crops ratio $s$.** Since ADCLR introduces the query crops, we further study the performance with different local and global crop ratios. Following iBOT (Zhou et al., 2022), we conduct the experiments by tweaking $s$, where $s$ is the scale dividing the local and global crops. The query crops and local crops are sampled from $(0.05, s)$. The global crops are sampled from $(0.4, 1)$. We fix the other hyper-parameter and pretrain ADCLR with 100 epochs. We report the linear classification accuracy in Table 6. We evaluate crop ratios smaller than 0.2 because of the low resolution of query crop With a large ratio, low-resolution crops may lose lots of local information.

Table 7: Ablation of the number of local views.

| crop strategy | 100 epochs | | 300 epochs | | #mem. |
|---|---|---|---|---|---|
| | Top-1 | time | Top-1 | time | |
| $2 \times 224^2$ | 68.9 | 16.1h | 73.9 | 47.6h | 9.8G |
| $2 \times 224^2 + 2 \times 96^2$ | 72.2 | 17.7h | 75.3 | 51.0h | 11.1G |
| $2 \times 224^2 + 6 \times 96^2$ | 74.6 | 20.8h | 76.7 | 60.9h | 13.4G |
| $2 \times 224^2 + 10 \times 96^2$ | **75.1** | 24.7h | **77.0** | 72.6h | 16.7G |

**Effect of local crops.** We further conduct ablation on two 8-GPU machines on the number of local views as reported in Table 7. We fix the batch size 1024 and evaluate it pretrained with 100 and 300 epochs. The multi-crop strategy follows SwAV (Caron et al., 2020), i.e., 2 global views (224*224) and 2, 6, 10 local views (96*96). Under the two-view setting, ADCLR achieves 68.9% top-1 accuracy with 100 epochs pretraining. With the same setting, we reproduce the result of DINO, resulting 67.7% top-1 accuracy with 100 epochs pretraining, where our ADCLR outperforms DINO **+1.2%** accuracy. For multi-crops setting, we run DINO with 300 epochs, using $2 \times 224^2$ + $10 \times 96^2$ crop strategy, resulting 76.2% (0.1% higher than reported in DINO) top-1 accuracy, ADCLR outperforms DINO by **+0.8%** top-1 accuracy. We find ADCLR gets higher accuracy under the two-crops strategy, which we guess using two global views for DINO is difficult to extract local information and the object in global views cannot be matched exactly. For ADCLR, the query tokens are cropped from the raw images with 16*16 pixels, which help capture accurate and dense information (see Fig. 1).

## 5 FURTHER DISCUSSION TO RELATED WORKS

We discuss the relation of ADCLR against similar approaches to better position our work.

**Relation to DINO (Caron et al., 2021). Connection.** ADCLR adopts the learning framework of DINO (EMA update, sharpening and centering). The global branch of ADCLR is the same as DINO. **Differences.** On the basis of DINO, ADCLR proposes a local branch to learn spatial-sensitive information. Besides, technically, ADCLR proposes the unidirectional cross attention mechanism.

**Relation to iBOT (Zhou et al., 2022). Connection.**. Both iBOT and ADCLR learn global and local information. **Differences.** iBOT learns the local information by adding MIM objective, which is now widely developed in MAE (He et al., 2021), SimMIM (Xie et al., 2021d). ADCLR learns the local information by maximizing the consistency of the same query crops in latent space (the variance is caused by the attention mechanism of two views). Compared with iBOT (masks patches), query tokens in ADCLR may be out of the views, which increases the difficulty of local contrasting.

**Relation to SelfPatch (Yun et al., 2022). Connection.** Both ADCLR and SelfPatch adopt DINO's framework on the patch level to learn dense information. **Differences.** The idea of SelfPatch is very straightforward, i.e., regard top-$k$ neighbors of query patch as positives, which may bring the noise (the selection of $k$). Besides, the local branch of SelfPatch requires input views has the same location, which decreases the learning difficulty. In contrast, ADCLR is simple yet also carefully designed. The local branch of ADCLR reserves the variance caused by augmentation (ADCLR can solve the positional shift caused by random resize and crop). Technically, SelfPatch designs the matching algorithm. For each query crop, it calculates the similarities and sorts them, which may bring a lot more complexity. ADCLR only introduces $Q$ query patches, where $1 \leq Q \leq 10$. The complexity brought by ADCLR can be ignored (ViT-B/16 and ViT-S/16 divide the image into 196 raw patches.)

## 6 CONCLUSION

This paper proposes ADCLR, a new contrastive learning method aiming to capture accurate and local information for vision representation. By introducing "query crop", ADCLR could easily match spatial location without complex operation, e.g., bi-linear interpolation. To make ADCLR more adaptable to downstream tasks and prevent collapse, we design unidirectional cross attention. Experiments on classification, detection, and segmentation show the effectiveness.

**Limitations.** Currently, the paradigm of "query crops" only supports low resolution (mapping the cropped region into small patches), which may lose semantic information. Hence, we think the multi-scale paradigm could help ADCLR a lot, which may be a good extension in future works.

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

## A   PROOF FOR THEOREM 1

**Proof** *For simplicity, we consider in one layer single-head self-attention block, and remove the $\mathbf{g}, \mathbf{b}$ re-scale parameters in layer normalization (Ba et al., 2016). Recall the self attention mechanism in Transformer, i.e., $Attn(\mathbf{x}, \mathbf{W}_Q, \mathbf{W}_K, \mathbf{W}_V) = Softmax(\frac{(\mathbf{x}\mathbf{W}_Q)(\mathbf{x}\mathbf{W}_K)^\top}{\sqrt{d}})\mathbf{x}\mathbf{W}_V$, where $\mathbf{W}_Q$, $\mathbf{W}_K$ and $\mathbf{W}_V$ are learnable weights, and $\sqrt{d}$ is the hidden dimension. Note that $\mathbf{x} \in \mathbb{R}^{(1+N+Q)\times d}$ includes $[CLS]$ token, raw patch sequence and query patch sequence. Let $s^{i;j} = \frac{(\mathbf{x}_i\mathbf{W}_Q)(\mathbf{x}_j\mathbf{W}_K)^\top}{\sqrt{d}}$, where $\frac{e^{-1}}{(N+Q)e+e^{-1}} \le s^{i;j} \le \frac{e}{(N+Q)e^{-1}+e}$, since the normalized $\|\mathbf{x}_i\mathbf{W}_Q\|_2^2 = 1$. Then, the input of two branches in contrastive learning becomes $[\mathbf{x}_A^0, \mathbf{x}_A^1, \mathbf{x}_A^2, \cdots, \mathbf{x}_A^Q, \cdots, \mathbf{x}_A^{N+Q}]$ for view A and $[\mathbf{x}_B^0, \mathbf{x}_B^1, \mathbf{x}_B^2, \cdots, \mathbf{x}_B^Q, \cdots, \mathbf{x}_B^{N+Q}]$ for view B. Note that we have $\mathbf{x}_A^i = \mathbf{x}_B^i$ for $1 \le i \le Q$. Then, the embedding of $i$-th patch $\mathbf{z}_A^i$ and $\mathbf{z}_B^i$ through self-attention block can be written as:*

$$\mathbf{z}_A^i = (\sum_{i=0}^{N+Q} s_A^{i;j} \cdot \mathbf{x}_A^j)\mathbf{W}_V, \quad \mathbf{z}_B^i = (\sum_{i=0}^{N+Q} s_B^{i;j} \cdot \mathbf{x}_B^j)\mathbf{W}_V, \quad 0 \le i \le N+Q \tag{7}$$

*Recall the objective of negative-free contrastive learning, i.e., aligning the representations $max \; sim(\mathbf{z}_A^i, \mathbf{z}_B^i)$ for $0 \le i \le Q$, where $\mathbf{z}_0$ is the embedding of $[CLS]$ token. For simplicity, we use the mean square error (MSE) objective for analysis. Take the gradient into Eq. 7, we get:*

$$\frac{\partial \mathcal{L}(z_A^i, z_B^i)}{\partial s_A^{i;j}} = \frac{\partial \mathcal{L}(z_A^i, z_B^i)}{\mathcal{H}(z_A^i, z_B^i)} \cdot \frac{\partial \mathcal{H}(z_A^i, z_B^i)}{\partial s_A^{i;j}} = \frac{\partial \|(\sum_{j=0}^{N+Q} s_A^{i;j}\mathbf{x}_A^j - \sum_{j=0}^{N+Q} s_A^{i;j}\mathbf{x}_B^j)\mathbf{W}_V\|_2^2}{\partial s_A^{i;j}} \tag{8}$$

*where $s_A^{i;j}$ means attention value of $i$-th patch to $j$-th patch in view A branch. Since $\mathbf{x}$ is standardized and $Q >> N$, Then, the numerator of Eq. 8 can be written as $\|(\sum_{j=0}^Q s_A^{i;j}\mathbf{x}_A^j - \sum_{j=0}^Q s_A^{i;j}\mathbf{x}_B^j)\mathbf{W}_V\|_2^2$. Then, the network could easily get collapse solution, i.e., making $s_A^{i;j} = s_B^{i;j}$ for $0 \le i, j \le Q$, leading the $\mathcal{L}(z_A^i, z_B^i) = 0$, since $\mathbf{x}_A^i = \mathbf{x}_B^i$ for $0 \le i \le Q$. Then, we complete the proof.*

## B   IMPLEMENTATIONS

**Backbone.** We use the standard ViT architecture (Dosovitskiy et al., 2020). It has several blocks composed of multi-head self-attention (MHSA) (Vaswani et al., 2017), MLP, and LayerNorm (Ba et al., 2016). In line with DINO and iBOT, we mainly adopt ViT-S/16 and ViT-B/16 as backbones.

**Pretraining hyper-parameters.** In line with DINO, we train with Adamw (Loshchilov & Hutter, 2018) and a batch size of 1024, distributed over 32 GPUs using ViT-S/16. The learning rate is linearly ramped up during the first 10 epochs to its base value determined with the following linear scaling rule (Chen et al., 2020a): lr = 0.0005, batchsize=256. After warmup, we decay the learning rate with a cosine schedule (Loshchilov & Hutter, 2016). The weight decay also follows a cosine scheduled from 0.04 to 0.4. The temperature $\tau$ is set to 0.1 while we use a linear warm-up for $\tau_t$ from 0.04 to 0.07 during the first 30 epochs. We follow the data augmentations of BYOL (Grill et al., 2020) (color jittering, Gaussian blur, and solarization) and multi-crop (Caron et al., 2020) with a bicubic interpolation to adapt the position embeddings to the scales. We set $\lambda = 0.5$ and query crop ratio from 0.05 to 0.15. The code with configuration details for reproducing will be publicly available.

Pytorch-liked code of unidirectional cross attention.

```
class Attention(nn.Module):
    def forward(self, cls, x, query):
        """
        :param cls: B, 1, C
        :param x: B, N, C
        :param query: B, Q, C
        """
        qkv = self.qkv(torch.cat([cls, x, query], dim=1)).reshape(B,
            1+N+Q, 3, self.num_heads, C // self.num_heads).permute(2, 0,
            3, 1, 4)
        q, k, v = qkv[0], qkv[1], qkv[2]
```

```
# cls and x forward
attn = (q[:, :, :1+N] @ k[:, :, :1+N].transpose(-2, -1)) *
    self.scale
attn = attn.softmax(dim=-1)
attn = self.attn_drop(attn)
x = (attn @ v[:, :, :1+N]).transpose(1, 2).reshape(B, N+1, C)

# query forward
attn = (q[:, :, N+1:] @ k[:, :, 1:N+1].transpose(-2, -1)) *
    self.scale
attn = attn.softmax(dim=-1)
attn = self.attn_drop(attn)
query = (attn @ v[:, :, 1:N+1]).transpose(1, 2).reshape(B, 1, C)
x = torch.cat([x, query], dim=1)
x = self.proj(x)
x = self.proj_drop(x)
return x[:, :1], x[:, 1:N+1], x[, N+1:], attn
```

