# OpenReview forum: "Patch-Level Contrasting without Patch Correspondence for Accurate and Dense Contrastive Representation Learning"
_ICLR.cc/2023/Conference — ICLR 2023 poster_

### Official Review · Reviewer_7W9J · 2022-10-23

**Confidence:** 2
**Clarity, Quality, Novelty And Reproducibility:** Some presentation is not clear.
**Correctness:** 3
**Technical Novelty And Significance:** 3
**Empirical Novelty And Significance:** 3
**Recommendation:** 6

**Strength And Weaknesses:**

--Strength

The paper introduces ADCLR, a novel patch-level contrastive learning method, improves the quality of dense representation of unsupervised learning. The experiments show the competitive performance of the proposed method. The paper provides detailed information for reproduction, and opens a new direction for future research.
The improvement is marginal comparing to iBOT on ImageNet.

--Weakness

Figure 1 does not describe the method well, e.g., what do we get after the augmentation image goes through Transformer Encoder? Reconstructed image or deep features?

In page 4, "...By feeding the whole sequence of two views xA and xB to..."  I cannot figure out how to get xA and XB.  It seems z is input, and goes through of attention layer, but where does x comes from? what is two views?

**Summary Of The Paper:**

This paper is about unsupervised learning of image classification. It is focused on patch-level contrastive by an attention mechanism. Experiments on ImageNet-1k, MSCOCO, ADE-20k verify the effectiveness.

**Summary Of The Review:**

The idea and experiments seem good, but the description of the method is not self-contain and clear.

---

> ### Author Response · Authors · 2022-11-06
> **Response to Reviewer 7W9J**
>
> Thank you for the time and thorough reviews. We feel sorry that our paper led to your confusion and misunderstandings.
>
> **Q1 (Fig. 1 is confusing)**. We get the embedding after taking the augmented views into transformer encoder and we have noted it in our revised version.
>
> **Q2 (input $X$)**. $X_a$ and $X_b$ are generated by feeding raw images $X$ into augmentation function twice.

---

### Official Review · Reviewer_D4eS · 2022-10-23

**Confidence:** 3
**Correctness:** 3
**Technical Novelty And Significance:** 3
**Empirical Novelty And Significance:** 3
**Recommendation:** 6

**Clarity, Quality, Novelty And Reproducibility:**

In general the quality of this work is good. The idea is clean. The experiments are solid.
The proposed method has some novelty in the transformer based patch-level contrastive learning.

**Strength And Weaknesses:**

Strengh
1. The framework is clean, simple, and effective.
2. This proposed method does not introduce too many extra parameters.
3. The method outperforms the previous methods.

Weakness
1. This paper claims that [CLS] token may extract mismatched information. However, the query patch token may also has this problem, if the patch does not exist in one augmented view.
2. How to select the query patches? Previous works like SoCo use selective search to select good patches. Selective search may also be used in this paper.

Other questions
1. In Table 7, are the local views used as raw patches or query patches?


**Summary Of The Paper:**

This paper focuses on learning dense contrastive representation based on vision transformer backbone. To achieve this goal, the authors propose a clean, simple and effective framework. In detail, they perform the cross-attention between query patches and raw patches. Then the updated query patch features are positive pairs. Despite the local contrastive loss, they also add global contrastive loss. The proposed method outperforms the previous methods.

**Summary Of The Review:**

I like the idea that perform attention between query patches and raw patches. Then contrast the query patch features.
I also like the analyses related to the collapse.
There are some details are not doing well, such as selecting the query patches.
In summary, I prefer to accept this paper.

---

> ### Author Response · Authors · 2022-11-06
> **Response to Reviewer D4eS**
>
> Thank you for the time, thorough reviews, and constructive suggestions, which inspire us a lot for future work. We are glad that you liked our idea.
>
> **Q1 (query patches do not exist in global views)**. As shown in Fig. 1 in our paper, the second query patch (white boat) does not exist in the top augmented views. However, through cross attention mechanism, transformer can pay attention to the region with similar contextual information. Note that the attention map is visualized by the model pretrained by ADCLR.
>
> **Q2 (How to select query patches)**. We just simply use the random crop for simplicity and efficiency. But we think the Selective Search could be a better approach to replace the random one. We would like to try it in our future work.
>
> **Q3 (query or raw patches)**. We only use global views as raw patches and the local views are independently fed into transformer. The loss between global and local views also follows DINO.

---

### Official Review · Reviewer_kB6d · 2022-10-26

**Confidence:** 3
**Correctness:** 4
**Technical Novelty And Significance:** 3
**Empirical Novelty And Significance:** 3
**Recommendation:** 8

**Clarity, Quality, Novelty And Reproducibility:**

There are two typos, on section 3.2 (i) the the same objects and (ii) tiken -> token. Otherwise the paper is very well written.

**Strength And Weaknesses:**

The first natural application is image classification, so ADCLR is benchmarked for that task. Common benchmarks in the field include (i) k-NN and (ii) top-K linear accuracy on imagenet and (iii) comparison against a strong supervised classifier on smaller datasets. The selection of methods (CNN and ViT) and datasets is comprehensive and shows  SOTA improvement.

However, the central application is dense prediction i.e. semantic segmentation and object detection, so ADCLR is also benchmarked for those tasks.


**Summary Of The Paper:**

This paper presents ADCLR, a method for contrastive and self supervised representation learning. The main target task is dense prediction i.e. semantic segmentation and object detection, however the framework fits the classification task as well.

Within the general framework of DINO Caron et al. (2021)  authors state their contributions to be (i) cross-views are used for contrasting learning, (ii) the unidirectional cross attention module, and (iii) achieving SOTA.


**Summary Of The Review:**

My recommendation is to accept this submission, the paper is sound and the results do support the claims.

---

> ### Author Response · Authors · 2022-11-06
> **Response to Reviewer kB6d**
>
> Thank you for the nice comments and valuable suggestions. We are encouraged that you appreciated our contributions including the novelty, soundness and solid experiments.
>
> **Q1 (typos)**. We have corrected the typos in our revised version.

---

### Official Review · Reviewer_wYqQ · 2022-10-27

**Confidence:** 3
**Clarity, Quality, Novelty And Reproducibility:** The paper is clearly written, and the…
**Correctness:** 3
**Technical Novelty And Significance:** 3
**Empirical Novelty And Significance:** 3
**Recommendation:** 6

**Strength And Weaknesses:**

Pros:
+ the idea of combing local and global contrastive learning is interesting, and also important for dense prediction tasks, as local and global contexts provide different semantic information of the image.
+ the results are strong.

Cons:
- As shown in Table 7, the local patch query introduces a lot of extra memory. It seems that the method sacrifices the memory overhead to gain performance. How about a larger-batch size? Would the method still be able to train with a reasonable memory consumption?
- The performance on ImageNet-1k seems to be minor compared to DINO and iBOT (only 0.3%). But from the comparison with state-of-the-arts on the dense prediction tasks, the gain is more clear. Could the authors explain the performance differences?


**Summary Of The Paper:**

This paper proposes a dense contrastive representation learning framework for visual feature learning in a self-supervised manner. The proposed framework considers a local patch query strategy for local contrasting together with a global contrasting. The method achieves superior performances on multiple challenging benchmarks for dense predictions.



**Summary Of The Review:**

The idea of performing local and global contrasting seems to be interesting, but on the other hand, this would also bring extra computational overhead, especially when the number of image crops is high. The authors need to further clarify their advantage on this point.

---

> ### Author Response · Authors · 2022-11-06
> **Response to Reviewer wYqQ**
>
> Thank you for the time and thorough reviews. We are glad that you liked our method and experiments. Here are our responses to your questions:
>
> **Q1 (extra memory)**. Conceptually, ADCLR only introduces several 16*16 patches (10), which is greatly smaller than original patches (196). We show the memory costs of other methods on our machine as follows:
>
> | Method | Crops | Memory |
> | -----       | -----     | ----------- |
> | DINO | 2 * 224$^2$ | 9.3G |
> | iBOT | 2 * 224$^2$ | 13.1G |
> | ADCLR (DINO) | 2 * 224$^2$ | 9.8G |
> | ADCLR (iBOT) | 2 * 224$^2$ | 14.6G |
> | DINO | 2 * 224$^2$ + 10 * 96$^2$ | 15.4G |
> | iBOT | 2 × 224$^2$ + 10 * 96$^2$ | 19.5G |
> | ADCLR (DINO) | 2 × 224$^2$ + 10 * 96$^2$ | 16.7G |
> | ADCLR (iBOT) | 2 × 224$^2$ + 10 * 96$^2$ | 20.8G |
>
> ADCLR only requires little extra memory on the basis of DINO. Besides, compared with iBOT, ADCLR (use DINO as baseline) spends less memory.
>
> **Q2 (downstream performance gain)**. As described in our first contribution, ADCLR is a patch-level contrastive learning method, which can learn more spatial-sensitive information than DINO. Therefore, ADCLR gets more significant improvement on dense prediction tasks. Similar phenomena can also be founded in DenseCL [1] v.s., Moco [2].
>
>
> [1] Wang X, Zhang R, Shen C, et al. Dense contrastive learning for self-supervised visual pre-training[C]//CVPR 2021.
>
> [2] He K, Fan H, Wu Y, et al. Momentum contrast for unsupervised visual representation learning[C]//CVPR 2020.

---

### Author Response · Authors · 2022-11-18
**Look forward to reply to our response and update**

Thanks to the efforts of ACs and all reviewers. Approaching the pdf updating DDL, is there anything needing added?

---

### Decision · Program_Chairs · 2023-01-20

**Decision:**

Accept: poster

**Justification For Why Not Higher Score:**

There are some concerns expressed by the review, and the idea behind the work is not groundbreaking so it may not be up to the oral or spotlight level.

**Justification For Why Not Lower Score:**

All reviewers support for acceptance so it should not be rejected.

**Metareview: Summary, Strengths And Weaknesses:**

The paper presents a dense contrastive representation learning framework for visual feature learning in a self-supervised manner. The proposed method combines local and global contrasting and achieves strong performance on multiple challenging benchmarks for dense predictions. The main strengths of the method are its clean, simple and effective framework and its ability to outperform previous methods. One weakness is that the local patch query strategy used in the method may introduce some extra memory and computation. The performance on ImageNet-1k is relatively minor compared to other benchmarks, such as DINO and iBOT. The authors could provide more explanation for this difference in performance. Overall, the paper is well-written and the results support the claims made by the authors.


**Note From Pc:**

if the above contains the word "oral" or "spotlight" please see: "oral" presentation means -> notable-top-5% and "spotlight" means -> notable-top-25%. As stated in our emails, we are disassociating presentation type from AC recommendations